# Targeting Peptides: The New Generation of Targeted Drug Delivery Systems

**DOI:** 10.3390/pharmaceutics15061648

**Published:** 2023-06-03

**Authors:** Biagio Todaro, Elisa Ottalagana, Stefano Luin, Melissa Santi

**Affiliations:** 1NEST Laboratory, Scuola Normale Superiore, Piazza San Silvestro 12, 56127 Pisa, Italy; elisa.ottalagana@sns.it (E.O.); s.luin@sns.it (S.L.); 2Fondazione Pisana per la Scienza, Via Ferruccio Giovannini 13, San Giuliano Terme, 56017 Pisa, Italy; 3NEST, Istituto Nanoscienze-CNR and Scuola Normale Superiore, Piazza San Silvestro 12, 56127 Pisa, Italy

**Keywords:** peptides, targeting, selection methods, chemical modifications, drug delivery systems

## Abstract

Peptides can act as targeting molecules, analogously to oligonucleotide aptamers and antibodies. They are particularly efficient in terms of production and stability in physiological environments; in recent years, they have been increasingly studied as targeting agents for several diseases, from tumors to central nervous system disorders, also thanks to the ability of some of them to cross the blood–brain barrier. In this review, we will describe the techniques employed for their experimental and in silico design, as well as their possible applications. We will also discuss advancements in their formulation and chemical modifications that make them even more stable and effective. Finally, we will discuss how their use could effectively help to overcome various physiological problems and improve existing treatments.

## 1. Introduction

The targeted delivery of drugs to a specific site in the body for the treatment of a disease is one of the greatest challenges in nanomedicine. The need to limit adverse side effects in patients led to the introduction of nanoparticles as more efficient drug delivery systems several decades ago [1]. Over the years, they have found many different applications, especially in the field of health care, with a particular focus on the treatment of tumors. Indeed, chemotherapeutic drugs are non-specific and cause several side effects, and there are currently no milder alternatives [1]. Nanoparticles can passively accumulate in tumors thanks to the Enhanced Permeability and Retention (EPR) effect, in virtue of the presence of damaged vessels and of an incomplete lymphatic system in the tumor microenvironment [2]. However, the EPR effect is not efficient enough to guarantee nanoparticles’ accumulation in a single specific site. An approach to overcome this problem is the use of targeting moieties on the nanoparticles surface, able to specifically recognize their targets, which should be present predominantly in the site of delivery [3]. Targeting molecules can be divided into four major classes: (i) antibodies, (ii) oligonucleotide aptamers, (iii) targeting peptides, and (iv) other molecules [4]. Antibodies show the highest affinity and specificity for their targets; however, they are very expensive and present some immunogenicity problems. Oligonucleotide aptamers and peptides have lower affinities but they have gained increasing attention due to their lower production costs and easier processes to obtain functional chemical modifications [4]. Moreover, they are usually more stable and less sensitive to proteases with respect to antibodies. Finally, one of the main advantages of peptides is that they are smaller molecules than antibodies and oligonucleotides, and therefore, the in silico selection processes are faster and more reliable [5]. Other molecules can be employed in nanoparticles functionalization such as serum proteins (e.g., transferrin or cytokines); however, their poor stability, high vulnerability to intracellular proteases, and the difficulty of obtaining the best orientation for targeting processes prevent their translation from the benchmark to the bedside [6]. Peptides are small amino acid chains that can be developed through experimental or in silico processes. The first mention of peptides as targeting molecules was given by Roger Brent [7]. His idea was based on the creation of a library of peptides constrained on a scaffold (usually the Thioredoxin A protein from *Escherichia coli*, TrxA) [7]. To do so, the sequence of each peptide is cloned in the sequence of the scaffold; thus, the resulting protein contains a peptide constrained at both ends. These peptides range in length from 25 to 35 amino acids to allow the formation of secondary structures that could increase the specificity for the target molecules. In recent decades, many other techniques have been developed to identify shorter peptides without the use of scaffolds [8,9]. In silico approaches are increasingly used to identify new molecules or to improve the chemico-physical features of those already identified by other techniques [10,11,12]. This allows reducing the experimental times and optimizing the selection processes, and therefore, a section of this review is dedicated to the analysis and description of the main computational models. Nowadays, many sequences have been identified for many targets, and several modifications and chemical approaches can be applied to improve their selection, binding affinity, stability, and specificity [13,14,15]. Nanoparticles decorated with targeting peptides are mainly used for the treatment of tumors. For example, they can be used for the specific delivery of nanoparticles to the tumor site, as described by Chanda et al., where the peptide Bombesin is used to deliver gold nanoparticles to different types of cancers [16]. Targeting peptides were employed in the functionalization of almost any type of nanoparticle (Figure 1) [17,18,19]. Due to their versatility and small size, peptides can be exploited alone or in combination with other peptides or different molecules to obtain drug delivery systems with multifunctional features [20]. Targeting peptides can also be used for the targeted delivery of small molecules, which can be either as drugs, as in the case of the work shown by Li and co-workers in the treatment of metastatic tumors with paclitaxel, or for imaging and diagnostics, such as in the work by Jean at. Al, in which specific molecular probes for mitochondrial imaging were developed [21,22]. In this review, we report the latest advances in the selection of targeting peptides, their chemical modifications, and their applications in the development of targeted drug delivery systems, and in each section, several examples of developed peptides are described.

## 2. Selection of Peptides

The fundamental feature of all targeting molecules and in particular of peptides is that they must be highly specific and affine for their targets. Indeed, the higher the specificity, the lower the possibility of having non-specific reactions and causing side effects. All methods available for targeting peptide selection are based on the screening of libraries, which contain thousands of random short amino acids sequences [23]. The selection processes follow common procedures, and they include (i) the identification of a target and the selection of a suitable library; (ii) a selection process consisting of several rounds in which specific peptide sequences are gradually selected; and (iii) the isolation and characterization of the most specific sequences [24]. After each cycle of selection, fewer and fewer peptide sequences are isolated until a pool of molecules is obtained that have high affinity and specificity for the target protein. Selection methods can be classified into experimental and in silico, depending on the use of laboratory processes or computer systems. In the following, we report a brief description of all experimental and in silico methods available for peptide selection, focusing our attention on innovations described in the literature in recent years (Figure 1).

### 2.1. Experimental Methods

Empirical methods directly use laboratory experiments for sequence selection and can be divided into in vitro and in vivo techniques [8]. In vitro protocols provide the selection starting from purified targets that are usually immobilized on a support and a chemically synthesized library [25]. These systems are easier and faster to use than the others but require a large amount of target protein and of the peptide library, and the method can be less efficient due to the uncontrolled position and orientation of the target protein immobilized on the support and the complications in producing adequate quantities of peptide sequences [23]. On the contrary, in vivo assays are based on the use of living organisms such as yeast, bacteria, or bacteriophages, which better mimic physiological environments [23]. Although they are very time consuming, these assays are preferred to the in vitro ones. In particular, two techniques are mostly used for peptide selection: two-hybrid assay and phage display assay.

#### 2.1.1. Two-Hybrid Systems

Two-hybrid systems are defined as “non-display” systems since the target protein and a single peptide sequence of the library are expressed in the same cell. The first two-hybrid system was based on the use of yeast as an incubator [26]. The main advantage of this technique is that the target protein does not have to be synthesized, as it is synthesized directly inside the yeast cell [26]. Yeast two-hybrid (Y2H) has been widely employed in recent years, and some targeting peptides have been identified with this method [27,28]. For example, in the work of Altwegg et al., a peptide able to specifically target and inhibit the activity of leucine-rich protein 1 (PELP1) was identified. PEPL1 is involved in breast cancer progression, and this peptide is able to block it in a very efficient way [28]. The Y2H system relies on the modular properties of many transcription factors (TFs) that are characterized by the presence of two distinct and independent parts, usually named the DNA-binding domain (BD) and the activation domain (AD). BD is commonly fused with the target protein and becomes the so called “bait”, while AD is cloned with the peptide library, becoming the “prey” [29,30]. DNA plasmids containing bait and prey are transformed in yeast cells and processed by their cell machinery. If the peptide and the protein interact with each other, the two domains, BD and AD, are close enough to promote the transcription of a reporter gene; this is usually a metabolic gene, which allows the growth of cells in a selective medium [31]. Yeast colonies obtained from the first round of selection are collected, and the plasmids with peptide sequences are isolated, amplified, and used for the second round of selection. Since each single colony obtained develops from a single clone, it will also have a single peptide sequence that will potentially have high affinity and selectivity for the protein of interest. The system was actually used mainly to identify protein–protein interactions for the identification of inhibitors and shows some major limitations due to the high numbers of false positives and negatives, the possibility of target protein toxicity for yeast, or the difficulty for the transformed plasmids to reach the synthetic machinery inside the cell nucleus [32]. Indeed, due to the high complexity of the system, sometimes the bait itself is able to activate the transcription of the reporter gene, without any interaction with a prey. Thus, the choice of the target protein is crucial to obtain a good selection process [30]. To limit this problem, yeast strains with different reporter genes are available, and in some cases, it is possible to use only a portion of the target protein [29]. However, mammalian proteins could be toxic for yeast cells or some post-translational modifications could be missed. For these reasons, other systems such as mammalian two-hybrid display or bacterial display, still based on the same concept, have been developed [33,34]. These systems allowed the identification not only of protein–protein interactions but also of DNA/RNA–protein interactions. Indeed, some of the problems often encountered with yeast cells, such as toxicity and post-translational modifications, were overcome with the mammalian two hybrid approach. However, the difficulty in understanding the results and the complexity of the systems has led to their being used little for peptides [33].

#### 2.1.2. Phage Display

Phage display represents the most used technique for the identification of targeting peptides. It is defined as in vivo since it relies on the use of bacteriophages, which are bacterial viruses that can be easily genetically engineered. The method uses peptide libraries fused within the genome of filamentous phages, such as M13 bacteriophages, to express the peptides fused with other capsid proteins. Different capsid proteins can be used for this purpose, and in particular, the pIII protein is the most exploited [8]. The importance of this method earned its inventors the Nobel Prize in 2018 [35]. Additionally, in this case, the phages, suitably modified to express peptide libraries, are used to identify the sequences most affine to a target protein in a process called bio-panning that, like the two-hybrid one, involves several rounds of selection and enrichment until a few highly related sequences are obtained [36]. When the solution with viral particles, each one having a single peptidic sequence fused with a capsid protein, is ready, it is put in contact with the target protein, which, in this case, is immobilized on a support. Only phages with the peptides related to the target protein remain attached, while the others are washed away [36]. These bounded phages are collected and amplified through bacterial infection. The “new” library is used for the following round of selection, until few sequences are obtained [36]. The technique is very laborious and time-consuming and requires the production of large amounts of the target protein, but most of the targeting peptides with diagnostic and targeting applications have been identified through this method [8,37,38,39]. For example, a 7-residue peptide was identified by Bakhshinejad and Sadeghizadeh using a commercial library, which was able to target colon carcinoma cells with no significant binding to normal cells [40]. Other peptides were identified to improve the binding affinity of antibodies for the treatment of SARS-CoV-2 variants and increase their neutralization potency [41]. Finally, in order to increase immune checkpoint inhibitors’ (ICIs) specificity, Li and co-workers improved the affinity of the LC4 peptide to create a system able to increase the ICIs’ tumor-targeting and reduce side effects, thereby increasing anti-cancer activity [42]. The widespread use of this technique led it to have many commercially available random peptide libraries from companies, with different characteristics such as the number of amino acid residues or different chemical modifications [40,43].

### 2.2. In Silico Methods

Although the two-hybrid and especially the phage display methods yield to peptides with high affinities for their targets, they are very laborious and time-consuming. Recently, in silico methods have been increasingly used to identify de novo peptides or optimize already existing ones. Then, molecules identified with these approaches are tested using in vitro and in vivo techniques. This was possible thanks to the availability of data obtained experimentally from the X-rays, NMR, and crystallography of proteins, peptides, and many other molecules. Information is stored in databases that can be used for the identification of protein–protein or protein-peptide interactions [44]. Generally, databases of peptides and proteins are grouped according to their biological functions or therapeutic effects; indeed, unique databases for antimicrobial peptides and targeting peptides or databases of disease-specific peptides have been created [45,46,47]. By conjugating these databases with different algorithms and tools, it is possible to identify or predict the tridimensional structures of peptides, their binding site with the targets, and also their binding energy and affinity [44]. The design of new peptide sequences can be carried out on the basis of the desired structure or sequence. The main available tools use the features of the amino acid sequence, from which we can obtain chemical–physical information of the peptide, which can then be optimized for stability, toxicity, and other parameters [47,48]. For example, ToxinPred2 and ProInflam are tools used for predicting peptides’ toxicity and pro-inflammatory antigenicity, respectively [49,50]. New and already existing peptides can be analyzed with different tools to identify secondary and tertiary structures and find the amino acids involved in the interaction with the specific protein. The modeling of peptides is carried out using different methods, which are classified into three main groups: (i) the homology-based approach, which relies on previously resolved structures; (ii) threading-based approaches, which rely on existing folds of proteins as the template; and (iii) the ab initio method, which exploits the chemico-physical properties of the peptide sequence to predict the structure with the lowest folding energy. One of the most used servers for the de novo prediction of peptide structure is PEP-FOLD (https://bioserv.rpbs.univ-paris-diderot.fr/services/PEP-FOLD4/ accessed on 31 May 2023) [51]. Interaction points, defined as hotspots, are extremely important in defining the ability of a peptide to bind specifically to a protein of interest. They were usually identified with the alanine substitution approach: single amino acid residues on the peptide are substituted with an alanine residue, which is neutral and less prone to interact with other residues. Binding energy and total energy are evaluated and compared with the wildtype sequence to understand which are the key points in the interaction between peptides and its target [52,53]. The use of molecular modeling approaches and virtual screening of peptidic libraries has recently led to the identification of many targeting peptides [11,54,55]. In the end, in silico methods allow work times and reduce research costs to be limited, and they have become a more and more fundamental step in the field of drug discovery not only for peptides but also for new drugs [56,57].

## 3. Chemical Modifications Designed to Increase Peptide’s Biological Properties

The chemical modification of peptides is an emerging research field in chemical biology, which allows the production of well-defined bioconjugates with improved chemical properties for biological studies and drug development [58,59]. Although peptides have been widely used as guide molecules for the targeted delivery of drugs, they possess several unfavorable features. Some peptides have a short half-life in plasma, largely due to enzymatic degradation by proteases. Sometimes, peptides may also have low affinity and selectivity for a target, or the passage through the cell membranes is limited due to solubility issues. Moreover, the detection and purification of peptides are limited by their size. To overcome these difficulties, several peptide modifications, such as amino acids, lipid or linker incorporation, addition, conjugation, or cyclization, have been used, e.g., to enhance the bioavailability and the pharmacokinetic properties, and they have become crucial means of performing biomedical imaging assays (Figure 2) [60,61]. Moreover, peptides that are generally quickly metabolized (e.g., in the liver) and excreted (e.g., from kidneys) need chemical modifications to extend their biological half-life and also to meet the imaging requirements of the radiolabeled peptides.

### 3.1. N- and C- Terminal Modifications

Targeting the N- or C- terminus of peptides is a promising strategy to neutralize the positive or negative charge at the free amino or carboxylic terminals, respectively, of a peptide [62]. Alkylation, acetylation, formylation, pyroglutamylation (pGlu), and biotinylation, along with conjugation with urea, carbamate, and sulfonamide, are the most used C- and N-terminal modifications [61,63,64,65,66]. These lead to the synthesis of molecules that more closely mimic the neutral state in the natural/native proteins and may increase the biological activity of the peptides [67]. Furthermore, N- and C- terminal modifications tend to increase the stability and prolong the peptide shelf life, enhancing the ability to resist enzymatic degradation. Peptides with neutrally charged ends are usually less soluble since the hydrogen bonds at the terminus are removed, and these kinds of chemical conjugations could change the conformation of the original peptide and reduce its bioactivity; these facts must be considered in the design of terminal-modified peptides [68,69]. An example of the success of a terminal modification is its application to Leu-enkephalin, a small endogenous peptide (Tyr-Gly-Gly-Phe-Leu) that acts as an agonist at both the μ- and δ-opioid receptors, found naturally in the brains of many animals, including humans. This peptide is rapidly degraded at the blood–brain barrier and in plasma. Thus, a suitable prodrug candidate containing N-terminal 4-imidazolidinone was proposed by Bak et al. for the delivery of Leu-enkephalin with increasing stability towards the enzymes [70].

### 3.2. Unnatural Amino Acid (UAA) Incorporation

Unnatural amino acid (UAA) incorporation is a great strategy to modify the properties of proteins, particularly to improve the pharmaceutical properties of bioactive peptides. Generally, UAA incorporation is commonly used to enhance the affinity and selectivity for a target and to increase the stability of peptides, as it generally induces or stabilizes secondary structures (α-helices, β-sheets, β-turns), increasing at the same time the resistance to proteases’ degradation [71,72,73,74]. Different strategies are available in this context (Figure 3), such as the following. (1) The incorporation of D-amino acids to increase resistance against a range of degradation enzymes: Peptides containing D-amino acids are significantly more stable, and sometimes more biologically active, than peptides containing only natural L-amino acids [75,76]. (2) Homo-amino acids: Compared to α-amino acids, homo-amino acids contain an extra CH_2_ or methylene group (homo). Depending on the position of the side chain on the 3-aminoalkanoic acid skeleton, β^2^- or β^3^ homo-amino acids (side chain on the α- or β- carbon) can be obtained: this addition may alter the biological activity or stability [77,78], for instance, increasing the in vivo half-life and/or the potency and reducing toxicity [79,80,81]. (3) N-methyl amino acids, which are amino acids that carrying a methyl group at the nitrogen instead of a proton: These amino acids are typically used to improve peptides stability and enhance intestinal permeability [82,83]. (4) α-methyl amino acids, in which the proton on the α-carbon atom is substituted by a methyl group: This modification makes the peptide as stable as possible to racemization/epimerization, increasing the proteolytic stability [84]. (5) Other unusual amino acids, such as citrulline, hydroxyproline, norleucine, 3-nitrotyrosine, nitroarginine, ornithine, naphtylalanine, methionine sulfoxide, or methionine sulfone: these amino acids are used to achieve higher enzymatic stability [85].

Selected examples are (i) the heptapeptide analogue of substance P, containing two N-methyl amino acids—these modifications confer almost complete resistance to degradation in the human brain [86]; (ii) the N-methylated analogs of endothelin, which revealed increased stability by 500–800 fold in comparison to its original form (half-life around 10–20 min) [87]; (iii) the N-methylation of neurotensin, which led to increased half-life in plasma with respect to its non-modified form [88]; (iv) the octapeptide octreotide, which was modified with the insertion of one D-amino acid at N-terminus and the amino-alcohol at C-terminus, and these modifications conferred an enhanced metabolic resistance, and thus the half-life was increased by about 1 h in the human body [89].

### 3.3. Peptide Biotinylation

Peptide biotinylation, imino-biotinylation, and desthio-biotinylation are efficient methods to label, detect, and purify peptides based on the very high affinity and binding specificity of biotin and biotin analogs for streptavidin (K_D_~10^−15^ M) [90]. Biotinylation can be performed either directly at the primary-terminal amino group or at the ε-amino group of a C-terminal lysine [91]. The extensive use of the biotin-streptavidin system has also identified the general need to include a linker or spacer longer than 4Å (~5 atoms) between the carboxy group of the biotin and the amino acid of the peptide sequence in order to avoid sterical hindrance and to ensure enough space for the biotin to reach the avidin binding pocket [92]. Moreover, the biotin–streptavidin binding is resistant to pH and/or temperature changes, to organic solutions, and to denaturing reagents [59].

### 3.4. Peptide Conjugation to Carrier Proteins

Peptide conjugation to carrier proteins is a well-known strategy to enhance peptides’ half-life and could also be exploited for direct targeting on a specific site [93,94,95]. For instance, albumin-conjugated peptides will take advantage of the long half-life of human albumin in plasma (about 19 days), its wide distribution, and its negligible immunogenic potential [96,97]. Conversely, peptide conjugation to carrier proteins may be a frequently necessary strategy to induce immune responses when injected into animals. Indeed, naked peptide antigens are often too small to create a successful generation of anti-peptide antibodies. Various carrier proteins, among which Keyhole Limpet Hemocyanin (KLH) and Bovine Serum Albumin (BSA), are the most common ones for weakly antigenic compounds, and different functional groups on the peptide, e.g., thiols, amines, and carboxylic acids groups, can be exploited to perform this conjugation, depending on the peptide chemistry, the features of the carrier proteins, and the intended use of the antibodies [59,98].

### 3.5. Lipid-Conjugated Peptides

The N-terminus of an amino acid or the side-chain of a lysine or cysteine can be conjugated with fatty acids, such as caprylic acid (C8), capric acid (C10), lauric acid (C12), myristic acid (C14), palmitic acid (C16), or stearic acid (C18). Peptides’ derivatization to lipids can be used for a number of different applications, e.g., (1) for improving peptide half-life in the circulation, mostly using long-chain fatty acids; (2) for increasing their antibacterial activity; (3) for raising the eukaryotic cell toxicity; and (4) for a faster passage through the cell membranes [99,100,101,102]. Moreover, derivatization of a peptide can also be performed with cholesterol via an N- or C-terminal inserted cysteine, especially in order to increase the potency of antiviral peptides, as an efficient gene delivery vector, for anticancer drug delivery, or when the peptide has a low in vivo half-life [103,104,105,106,107].

### 3.6. Cyclization

Another strategy is the creation of an arrangement representative of a protein secondary structure, e.g., creating peptide loops. Constraining peptides via cyclization is the main approach to achieve this strategy, which can play a pivotal role in several biological functions, allowing enhanced receptor selectivity, potency, and thermo-stability, as well as more proteolytic resistance compared to their linear analogs [68,95]. An example is the selective and potent α_v_β_3_, α_v_β_1_ and α_v_β_5_ integrin receptors inhibitor c(RGDfV) peptide, with IC_50_ values of 89, 335, and 440 nM, respectively. RGD is the minimal recognition sequence for integrin binding found in many extracellular matrix (ECM) and serum proteins. The presence of the D-Phe at the position 4 is essential for the preservation of the α_v_β_3_-binding affinity. The analog cilengitide contains methylated D-Phe (RGDf(NMe)V), which results in a more active and stable peptide in the human body (with a half-life of about four hours) than the natural compound [108]. Cyclized peptides are generally prepared via head-to-tail cyclization or side-chain-to-side-chain cyclization. The most frequently used peptide cyclization methods are Cys–Cys cyclization and amide cyclization [109]. The former consists of the formation of a disulfide (S-S) bond between the thiol side chains of two cysteine residues in a peptide, e.g., in disulfide-rich peptides (DSRs) and cyclotides, whereas the latter is the direct condensation between terminal carboxylic acid and amine groups. Amide cyclization leads to chemically more stable bonds over disulfides. In both cases, the syntheses are very challenging due to the need to avoid dimerization, racemization, peptide capping, and other undesired side reactions, which could be favored for entropic reasons [110,111]. Therefore, it is very important to perform the synthetic processes under high-dilution environments and at specific temperatures and catalysis conditions, or to apply different series of amino-acid-protective groups using orthogonal protective group strategies, in order to be able to ensure product formation as a unique and single topological isomer [61,112]. Merging approaches aim to fulfill these challenges by performing these synthetic processes using small organic molecule scaffolds, enzymatic methods, or protein tags [113]. For instance, CLIPS (Chemical Linkage of Peptides onto Scaffolds) technology used for peptide cyclization is a widespread new generation method for large phage-displayed libraries of thioether-bridged monocyclic, bicyclic, tricyclic, and tetracyclic peptides [114,115,116].

### 3.7. Addition of Linkers and Spacers

Linkers and spacers are flexible molecules that can be incorporated anywhere in the peptide. In general, linkers are mainly used for enhancing the stability, proteolytic resistance, and solubility of peptides. The addition of a linker is also commonly used to reduce the immunogenicity of peptides and to produce a distance between the peptide and the cargo. In this case, the cargo is represented by a drug, dye, tag, a carrier protein, or other molecules of interest, and the linker has a pivotal role in reducing steric hindrance at the binding sites [61,117]. Linker addition is generally made via standard amide, thiol-maleimide, oxime ligation, or click chemistry, depending on the peptide sequences and the linkers’ chemical nature. In literature, there are several examples of peptide-conjugating linkers, both rigid, such as Beta-alanine, 4-aminobutyric acid, (2-aminoethoxy) acetic acid, 5-aminovaleric acid, 6-aminohexanoic acid, or Trioxatridecan-succinamic acid, and more flexible such as poly-lysine or Polyethylene glycol (PEG) [118]. PEG is a class of molecules with different molecular weights, and it is one of the most employed linkers. They are versatile, hydrophilic, nonionic, biocompatible, and FDA-approved linkers. Depending on the length and molecular weight, there are monodispersed PEGs, composed of a precisely defined number of PEG units (fewer than 60 atoms in length, with MWs ranging from 5 to 40 kDa), and polydispersed PEGs, which are mixtures of different oligomers with an average length. Peptides’ PEGylation is widely used for increasing the solubility and to prolong their half-lives in vivo. Indeed, the PEGylation process totally reduces the clearance of small peptides by circumventing the glomerular filtration and protects the peptide from proteolytic degradation by forming a protective shell around the molecule [119]. In addition to this, PEG greatly reduces the non-specific adsorption of serum proteins. Indeed, for instance, it is usually coupled with nanoparticles to significantly decrease the formation of the protein corona, leading to a rise in targeting capabilities and to a higher half-life in the plasma of the nanosystem [120,121,122,123]. Furthermore, PEGs are typically employed to mask antigenic sites, thus reducing immune recognition of antigens and limiting immune responses in the host [124]. There are two major disadvantages of conjugation with PEG polymers: on the one hand, the conjugation may reduce the peptide-binding affinity and potency compared with the original therapeutic; on the other hand, the fact that PEGylated peptides are not fully biodegradable may result in toxic accumulation in kidneys [120,125,126,127].

### 3.8. Metal Chelators/Radioligands

The development of target-specific radiopharmaceuticals based on radiometals is a growing field of research for non-invasive disease detection and cancer radiotherapy. Radiometals for diagnostic imaging are mainly used in two radio imaging modalities: single photon emission computed tomography (SPECT) and positron emission tomography (PET). PET scans offer much better image resolution than SPECT scans, but they are more expensive because of the need to produce, in a cyclotron, short-lived PET radionuclides (^18^F, ^11^C, ^13^N, ^15^O). On the contrary, the use of low-energy gamma-emitting radiometals (mainly ^111^In, ^90^Y, ^99m^Tc, ^177^Lu, ^68^Ga) used in the design of SPECT radiopharmaceuticals within a targeting peptide structure has become an indispensable tool in current imaging techniques and therefore in diagnostics [128,129]. Radiometals into peptides are generally incorporated through a suitable bifunctional chelate, which depends on the oxidation state of the radiometal, and it can hold it tightly while forming a kinetically inert conjugation with the active groups of the biomolecule. The complex must be thermodynamically stable to prevent the radiometal dissociation from its chelator, which results in the possible accumulation of radioactivity in non-target organs (bringing it to non-specific signals and to radiation toxicity) [130]. Based on their excellent in vivo properties, several bifunctional chelators, typically containing a macrocyclic core such as DOTA, DTPA, NOTA, and TETA, have been labeled with radiometals and used in diagnostic imaging. The covalent conjugation to the peptide is frequently achieved at the N-terminus or at the C-terminus via a C-terminal lysine, allowing very thermodynamically stable and kinetically inert complexes [63,131,132,133].

### 3.9. Stable-Isotope-Labeled (SIL) Peptides

Stable-isotope-labeled (SIL) peptides are biomolecules composed of a heavy-isotope (^2^H, ^13^C, ^15^N)-variant amino acid sequence. The tiny mass differences with their non-labeled counterparts make SIL peptides extremely important in proteomic analysis, especially at very low concentrations, while retaining equal physiochemical properties and chemical reactivity [134]. Indeed, SIL peptides are generally employed as internal or surrogate standards in liquid chromatography for profiling transcriptomic proteins, although the production of SIL-proteins can be limited by cost and commercial availability [135,136,137].

### 3.10. Fluorescently Labeled Peptides

Recently, both fluorescence instrumentation and the synthesis of new fluorophores have allowed outstanding progress toward biomolecule detection based on fluorescence techniques. Fluorescently labeled peptides are frequently used for pre-clinical research, especially in in vitro/vivo fluorescence-based assays, such as protein–protein interaction analyses, flow cytometry, and localization studies [138,139,140]. For example, the 18F-labeled vasoactive intestinal peptide (VIP) analogue, a gut neuroendocrine mediator, showed high stability, receptor specificity, and fast elimination in preliminary studies conducted in mice [141]. The conception of Fluorescence Resonance Energy Transfer (FRET), in particular the idea of using other chromophores as fluorophores quenchers, has led towards the development of quenched fluorescent peptides or FRET peptides. By relying on the FRET concept, the “distance-dependent” interaction can be monitored: only at low distances (below some nanometers) does the quencher efficiently block the emission of the fluorophore. For instance, Carmona et al. employed FRET peptides containing ortho-aminobenzoic acid (Abz) as a fluorescent group and 2, 4-dinitrophenyl (2,-DNP) as a quencer, showing that hydrolysis of a “FRET peptide” bond between the donor and acceptor pair generates fluorescence that permits the measurement of the activity of nanomolar concentrations of an enzyme [142]. More recently, Bernegger et al. designed and employed a FRET peptide, harboring 2-aminobenzoyl (2-Abz) as fluorophore and 3-nitro-tyrosine Y(NO_2_) as a quencher, for the efficient development of novel metal ion-dependent protease inhibitors, which might help to protect the gastric epithelial integrity and fight Helicobacter pylori infections [143]. Despite the various drawbacks, including autofluorescence, detector, and optical noise, as well as a low signal-to-noise ratio, this strategy is very useful for detecting the fluorescence signal continuously and to measure changes in distance between molecules [144,145].

### 3.11. Conjugation to Antibodies

The conjugation of peptides with antibodies or antibody portions, such as the fragment crystallizable region (Fc) or the fragment antigen-binding region (Fab), allows the generation of smart therapeutic agents with the potential to be both more effective and to have fewer side effects of the single counterparts. The resulting biomolecule will usually possess both the biologic actions of the peptide and the long half-life of the antibody domain. Moreover, the conjugates that used the antibodies portion are expected to have higher tissue penetration as compared to the whole antibodies, given their small sizes. In addition to their potential pharmacokinetic advantages, the production and manufacturing of peptide-Fc/Fab fusions can be more efficient and less time-consuming than those for full-length monoclonal antibodies (mAbs) [146,147]. These complexes are an attractive therapeutic modality for medical treatment, as they allow the increase in the treatment response rate and prevent severe side effects. For instance, J. E. Lachowicz’s group developed a novel Brain-Penetrant Peptide–mAb Conjugate to target human epidermal growth factor receptor2 (HER2)-positive intracranial cancer cells in vivo; the conjugation of the anti-HER2 monoclonal antibody, which known to be a potent therapeutic against breast cancer but ineffective against brain metastases due to poor brain penetration, to the angiopep-2 peptide, which is able to cross the blood–brain barrier (BBB) via receptor-mediated transcytosis, conferred properties of increased uptake in brain endothelial cells as well as BBB permeability [148]. More recently, researchers from Janssen Biotherapeutics (USA) designed a novel mAb-peptide conjugate, in which a bromoacetylated peptide was conjugated to a mAb cysteine residue, which resulted in more potency and affinity to glucagon and glucagon-like peptide 1 receptors in vivo than the single counterparts [149]. One of the major challenges with regard to process consistency and product characterization is that antibodies usually carry several copies of lysines and cysteines exposed at their surface; thus, the ligation of the peptide usually results in a highly heterogeneous and random mixture, affecting the efficacy, safety, pharmacokinetics, and immunogenicity of the conjugated antibody. To overcome this issue, some template-directed ligations were emerged as powerful means to regioselectively label both prior modified or unmodified antibodies mAb, using genetic engineering or chemical modification processes [150,151].

### 3.12. Post-Translational Protein Modifications (PTMs)

Protein posttranslational modifications (PTMs) are several different processes through which proteins size, charge, structure, and conformation are altered. Multicellular eukaryotic organisms rely on several PTMs affecting essential roles in many cellular processes such as gene expression, signal transduction in plants and animals, the mediation of protein compartmentalization, sequestration, degradation, elimination, and interactions with other proteins, enhancing the cell-penetrating ability of peptide [151,152]. The most common PTMs are peptides’/proteins’ sulfation and phosphorylation (the addition of a covalently bound sulfate group or phosphate group, respectively, usually on the phenolic side chain of tyrosine residues); methylation, which involves the addition of a methyl group typically to amino acids residues, especially cysteine (mainly used as an anti-oxidative strategy and for the identification and characterization of proteins by mass spectrometry); and prenylation, which is the addition of an isoprenoid moiety (typically three or four isoprene units) at the C-terminus of amino acids, mainly to peptides’ cysteine residues. However, a great summary of PTMs is reported elsewhere [153].

## 4. Peptide-Based Targeting Strategies for Drug Delivery Systems

### 4.1. Peptide–Drug Conjugates (PDCs)

Peptide–drug conjugates (PDCs) are a type of targeted therapeutic in which a peptide is covalently bonded to a small organic drug molecule. PDC is usually made up of three parts: a homing peptide, a linker, and a payload [154,155]. The functional groups in the linker region determine different types of drug release and can be classified into four different classes: enzyme-cleavable (ester, amide, and carbamate), acid-cleavable (hydrazone and carbonate), reducible disulfide, and non-cleavable (thioether, oxime, and triazole). This classification depends on the behavior of these functional groups after cellular uptake. For instance, enzyme-cleavable ester or amide bonds can be selectively cleaved in the tumor microenvironment or in the lysosomes, which are rich in esterases and amidases. Acid-cleavable linkers have also gained attention because they are cut primarily in the acidic tumor microenvironment while maintaining stability in blood circulation. Since tumor cells are also characterized by an extremely reductive environment, reducible disulfide linkers have also been developed in order to allow drug release only in those cells. On the contrary, the rationale behind the use of non-cleavable linkers is based on the fact that some drugs should not be prematurely released in plasma before reaching the target cancer site. Generally, it is important that the PDCs are stable enough to avoid degradation before reaching the target where the drug must be released [156]. PDCs are similar to antibody–drug conjugates (ADCs); the only difference is the use of a homing peptide instead of an antibody. However, many monoclonal antibodies (mAbs) are unable to penetrate tumors due to their size and can be immunogenic, leading to accumulation in organs such as the liver and kidneys. Additionally, mAbs are expensive and time-consuming to produce [157]. Most of these drawbacks can be overcome with PDCs, which are cheaper, easier to produce, and less immunogenic [158]. Although they have a short half-life and are rapidly eliminated by the kidneys [159], several modifications and stabilizations of the peptide side chains and backbone have been developed to improve pharmacodynamic properties [159,160]. PDCs have been widely studied, especially in the treatment of tumors and bacterial infections. For example, DTX-P7, a PDC composed of docetaxel (DTX) and a heptapeptide (P7), is able to bind Hsp90, an upregulated protein on the surface of lung cancer cells [161], and to induce tumor cell apoptosis, making it a promising cancer therapy [162]. Another example is ANG1005, a peptide–drug conjugate consisting of three paclitaxel molecules linked to the peptide Angiopep-2, which can cross the blood–brain barrier and reach the central nervous system (CNS). Kumthekar and coworkers demonstrated that treatment with ANG1005 increased the survival of patients with leptomeningeal metastasis by 8 months compared to the median survival time of 4 months with the standard treatment and 2 months without treatment [21]. BT8009, designed by Mudd et al., is a bicycle toxin conjugate that targets Nectin-4 cell adhesion molecules, which are overexpressed in several tumor types, such as ovarian [163], lung [164], and esophageal cancers [165]; BT8009 is currently being evaluated clinically, after demonstrating potent antitumor activity in mouse models [166]. DCs can also be used for antimicrobial activity and have been shown to overcome bacterial resistance in some cases. For example, conjugating vancomycin with polycationic peptides appears to be a promising strategy for combating multidrug-resistant bacteria [167,168]. The cationic charge of the peptide allows selectivity for negatively charged microbial cytoplasmic membranes over zwitterionic mammalian membranes [169].

### 4.2. Liposomes

Liposomes are small, artificial, spherical vesicles that are obtained through the self-assembly of amphiphilic lipid molecules in a solution. These molecules can be made of both natural (phosphatidylcholine, phosphatidylethanolamine, phosphatidylserine, phosphatidylinositol, phosphatidylglecerol, and phosphatidic acid) and synthetic (palmitic and stearic acid-based synthetic phospholipids) phospholipids [170]. The liposomal membrane consists of one or more lipid bilayers surrounding aqueous units, where the hydrophilic heads are oriented towards the interior and exterior aqueous phases [171]. This structural organization allows liposomes to be used as nanocarriers for different types of molecules. For example, hydrophilic molecules can be loaded in the internal aqueous core, hydrophobic molecules in the lipid bilayer, and amphiphilic molecules at the interface between the water and lipid bilayer [172]. The liposome features, such as particle size, rigidity, fluidity, stability, and electrical charge, as well as the interaction with cell membranes, are strongly associated with the lipids composition. Different types of interactions have been observed, including endocytosis, local fusion, phagocytosis, and absorption into the cell membrane [170]. Decorating liposomes with external peptides is a common technique to confer or increase targeting capacities [173]. For example, D’Avanzo et al. recently developed a nanovesicle loaded with two chemotherapy drugs, and decorated it on the surface with LinTT1 peptide motifs to target the p32 protein, which is overexpressed in several types of malignancies, including colorectal [174], prostate [175], and breast cancers [176]. The resultant liposomes showed enhanced anticancer activity in positive estrogen receptor (MCF-7) and triple-negative breast cancer (MDA-MB-231) cells compared to single chemotherapy drugs [177]. Shen et al. synthesized liposomes encapsulating a peptide derived from l-lactate dehydrogenase A chain (PDBSN) and conjugated with a visceral-fat targeting ligand and a cell-penetrating peptide (CPP). The authors showed how this nanosystem is able to reduce adipose mass and improve glucose metabolism, as well as lipid homeostasis, in an obese mouse model induced by a high-fat diet (HFD). Moreover, these liposomes seemed to reactivate the AMPK signal, which is impaired by the HFD [178].

### 4.3. Polymeric Micelles (PMs)

Polymeric micelles (PMs) are smaller, more monodispersed, and more stable than lipid vesicles. They are synthesized by self-assembly in aqueous solution from amphiphilic block copolymers, such as polystyrene and polyethylene glycol, or triblock copolymers [179], such as poloxamers. PMs can also be produced from graft copolymers [180], such as poly(lactic-co-glycolic acid) (PLGA) [181,182], chitosan, and ionic copolymers [183], such as poly(ethylene glycol)-poly(ε-caprolactone)-g-polyethyleneimine [184]. All these molecules and the hydrophobic core can act as a reservoir for hydrophobic drugs, therefore making PMs ideal candidates for drug delivery. This is especially true for anticancer drugs, which have low water solubility due to their polycyclic structure and a rapid metabolic degradation, which limits their clinical usage [185]. Moreover, the small size of PMs allows for improved accumulation in tumors through the EPR effect [186,187]. In 2017, Quader et al. designed cRGD-decorated epirubicin-loaded polymeric micelles (cRGD-Epi/m) that effectively suppressed the growth of an orthotopic glioblastoma (GBM) model [188]. Moreover, the cyclic RGD is more stable and more active than their linear counterparts [189]. More recently, Chen et al. developed pH-sensitive PMs functionalized with spermine (SPM), a tumor-binding ligand, and loaded with hydrophobic paclitaxel (PTX). These PMs showed an improved therapeutic effect compared to free PTX, due to an enhanced intracellular uptake and a burst drug release [190]. Tang et al. developed pH-sensitive PMs, called TX/PMs-HE-CPP, loaded with Taxol (TX) and conjugated to a cell-penetrating peptide (RACPP), for tumor-targeted delivery. These PMs exhibited improved anti-tumor efficacy compared to unmodified PMs and Taxol [191].

### 4.4. Iron Oxide Nanoparticles (IONPs)

Iron oxide nanoparticles (IONPs) belong to the ferrimagnetic class of magnetic materials and can be synthesized using physical (deposition of gas phase and electron beam lithography) [192], chemical (e.g., sol-gel method, oxidation, and chemical coprecipitation) [193], or biological (microbial incubation) [194] methods. Among these, chemical techniques are favored due to their lower cost and their ability to provide high yields [195]. Ferromagnetic-material NPs with a size of 10–20 nm exhibit a form of magnetism known as superparamagnetism. There are different types of iron oxide-based nanoparticles, including the ones made of magnetite (Fe_3_O_4_), maghemite (γ-Fe_2_O_3_), and mixed ferrites [196]. Surface coating with various molecules, such as polymeric, inorganic, or organic materials, such as peptides, can improve the stability, biocompatibility, and solubility of IONPs, thereby expanding their range of applications [197]. Liao et al. achieved in vivo contrast-enhanced MRI using ultrasmall superparamagnetic iron oxide nanoparticles (USPIONs) coated with different peptides. Among these, 2PG-S*VVVT-PEG4-ol-coated USPIONs demonstrated the best magnetic resonance properties and highest biocompatibility, better than commercially available MRI contrast agents. 2PG-S*VVVT is a bisphosphorylated peptide with a 2-phosphoglycolic acid (2PG) and a Ser(PO_3_H_2_) group (S*) [198]. A different application of IONPs was reported by Ferreras et al., who developed iron oxide superparamagnetic nanoparticles (MNPs) by combining in the coating FLR peptides, which allow targeting and internalization in cells, and plasmid (p)DNA. This nanoparticle system was able to significantly increase reporter gene expression in NIH-3T3 cells after short incubations in the presence of a magnetic field, compared to either magnetic field or FLR-DNA nanocomplexes alone [199].

### 4.5. Gold Nanoparticles (GNPs)

Gold nanoparticles (GNPs) have gained attention due to their stability and ease of synthesis. The optical behavior of GNPs is dependent on their surface plasmon resonance (SPR), which is a collective oscillation of free electrons that create oscillating positive and negative surface charges [200]. Conventional Raman spectroscopy and surface-enhanced Raman spectroscopy (SERS) have also been exploited to determine the composition, the properties, and the structural arrangement of GNPs and other nanomaterials under analysis [201,202]. SPR occurs in the visible and infrared regions of the spectrum, depending strongly on both the size and the shape of the GNPs. Moreover, GNPs can be conjugated with various molecules, making them useful for a range of medical applications, such as bio-imaging, gene delivery, contrast enhancement for X-ray computed tomography, targeted drug delivery, diagnostics, plasmonic bio-sensing, colorimetric sensing, tissue engineering, photo-induced therapy, and cancer therapy [203]. In particular, peptides can be linked to the gold surface thanks to the strong gold–thiol interaction exploiting a (usually N- or C-terminal) cysteine. An example of such an application is the use of peptide-coated GNPs to treat acute lung injury progression, as reported by Wang et al. These nanoparticles induce M2 macrophage polarization in vitro and in vivo, effectively regulating lung inflammation, protecting the lung from injury, and promoting inflammation resolution [204]. De Alteriis et al. developed another type of nano-system, AuNPs coated with indolicidin, a cathelicidin-related antimicrobial peptide, to treat Candida albicans biofilms. The treatment with AuNPs-indolicidin was able to inhibit the formation of biofilms and impair preformed mature biofilms [205]. More recently, D. Perrins et al. functionalized ultrasmall gold nanoparticles with the cRGD peptide, which binds to the α_V_β_3_ integrin receptor, which is overexpressed in many tumors such as melanoma [76,206]. Moreover, cRGD-ultrasmall GNPs loaded with the maytansinoid drug (DM1), a tubulin polymerization inhibitor, showed increased cytotoxicity selectively on cells that expressed the α_V_β_3_ integrin [207,208].

### 4.6. Quantum Dots (QDs)

Quantum dots are nanometer-sized semiconductor crystals ranging in size from 2 to 10 nm. Synthesis techniques for QDs are divided into “top-down” and “bottom-up” approaches. Both can also be classified into physical (without chemical transformation of the matter), chemical (through chemical reactions), and biological (through living organisms) methods. The most widely used synthesis approaches are the organometallic bottom-up synthesis, which consists of three steps carried out in organic solvents under a high-temperature and inert environment, and the aqueous solution-based synthesis, which is more environmentally safe. In both synthesis routes, the size and emission wavelength of QDs can be monitored by controlling the reaction conditions [209]. The structure of QDs consists of an inorganic core, with unique optical properties and semiconductor features, often surrounded by a shell of a different semiconductor. The QD can also be capped to improve solubility in aqueous buffers and/or for linking other moieties for various functionalizations. Moreover, several ligands can be used in the synthesis of QDs to facilitate bioconjugation with carbohydrates, DNA fragments, and peptides [210]. QDs are attractive due to their quite narrow-spectrum photoluminescent emission (PL), high extinction coefficient, high fluorescence quantum yield, high photobleaching stability, and blinking [210]. Brunetti et al. developed near-infrared quantum dots (NIR-QDs) functionalized with cancer-selective tetra-branched peptides (NT4-QDs) for in vivo imaging [209,211]. They observed the specific uptake of NT4-QDs in human cancer cells in vitro and a much higher selective accumulation and retention of targeted QDs at the tumor site, with respect to non-targeted QDs, in a colon cancer mouse model. Furthermore, in 2022, Jin et al. reported a QD-based Förster Resonance Energy Transfer (FRET) technique for cancer diagnosis. This consists in synthesizing a FRET sensor through covalent peptide bond formation, which can detect the cancer biomarker MMP-14 with high sensitivity [212]. QDs are coated with dithiolane poly(ethylene glycol) (PEG) ligands bearing a terminal carboxyl group that allows amide bond formation with specific TAMRA-labeled peptides, which are a substrate sequence for MMP-14. The TAMRA absorption profile exhibits a pronounced spectral overlap with the photoluminescence (PL) of green-emitting QDs, making this donor−acceptor pair ideal for FRET. At the beginning, the PL of the excited QDs is low due to FRET but increases upon incubation with MMP-14 due to peptide cleavage and loss of FRET interactions. Very recently, Haider et al. developed QD-Ps, which are graphene oxide QDs functionalized with a peptide (GILGFVFTL) that targets the placenta-specific protein 1 (PLAC-1), which is overexpressed in colorectal cancer (CRC). The prepared QD-P demonstrated a significant increase in targeting and a specific uptake in PLAC-1-expressing cells compared to non-functionalized QD [213].

### 4.7. Mesoporous Silica Nanoparticles (MSNs)

Mesoporous silica nanoparticles (MSNs) are composed of colloidal amorphous silicon oxide. The majority of MSNs are produced using the method of Stöber, a pioneer who developed a system of chemical reactions for the synthesis of spherical monodisperse micron-sized silica particles [214]. Stöber’s method is a so-called sol-gel process, which consists of hydrolysis and condensation of alkoxide monomers into a colloidal solution (sol), which represents the precursor to form an ordered network (gel) of polymers or discrete particles. Over the years, Stober’s method has undergone many modifications in order to obtain monodisperse, ordered, and nanosized silica particles [215,216]. MSNs have specific and controllable physico-chemical properties, including the presence of internal and/or external pores, uniform and tunable pore size, easy surface functionalization, and possible gating mechanisms for pore closure and opening. Pores’ features make MSNs interesting nanocarriers for controlled drug delivery [215,216]. Compared to other nanovectors, MSNs are more stable to degradation and mechanical stress than liposomes; therefore, they do not require external stabilization during synthesis processes [217]. However, reproducibility at the industrial scale is critical, as is the loading capacity of MSNs, since not all drugs can be incorporated at an appropriate concentration; this critically determines the total concentration of NPs that need to be administered to achieve a therapeutic effect [218]. For example, Yan et al. developed MSNs, loaded with Paclitaxel (PTX), an antimitotic and anticancer drug clinically used to treat solid tumors, and double-tagged with folic acid (FA) and arginine-glycine-aspartic acid (RGD) peptides to target folic acid receptors (FR) and αvβ_3_ integrins; these nanoparticles showed almost twice the inhibitory efficacy of free PTX, demonstrating their potential as nanocarriers for targeting breast cancer cells [219]. Tenland et al. reported an interesting study on the use of MSNs to target Mycobacterium tuberculosis: they prepared MSNs loaded with the antimicrobial peptide NZX to treat multi-drug-resistant strains of M. tuberculosis. Loading NZX into MSNs increased the inhibition of intracellular mycobacteria in primary macrophages and maintained the ability to eradicate M. tuberculosis in vivo in a murine model [220].

## 5. Conclusions

Peptides are defined as small molecules composed of approximately 50 or fewer amino acid residues. Many peptides are currently used as drugs, such as insulin, which was identified a few decades ago and whose discovery was one of the most important from a scientific and human healthcare point of view. Targeting peptides, in particular, are a subclass of molecules that have been considered unsuitable for therapies for a long time due to their low-affinity constants against their specific targets. However, the difficulties and costs of producing other molecules such as antibodies have led researchers to reevaluate peptides. Antibodies were always considered the elite class of molecules for targeting due to their high affinity and specificity for their targets. However, their complex structure, the high cost of production, and, above all, the problems of immunogenicity made them less attractive. On the contrary, peptides showed very interesting features since their production has lower costs, they are very small and therefore less susceptible to proteases, and finally they can be chemically modified without too much effort and with high yields. The development of new techniques for their identification made their use possible in various applications. Although the experimental techniques are very laborious, some variants have been developed that can reduce selection times and production costs. In this review, we discussed the advantages and drawbacks of available experimental methods that were used to reduce the selection times and production costs. The advancement of digital technologies and the ever-increasing computing power of modern computers made the selection of targeting molecules much faster and cheaper through in silico approaches. It is increasingly easier to be able to recreate digitally complex biological systems, and there are numerous approaches to study the possible outputs deriving from interactions between molecules. We have discussed computational approaches since we strongly believe that this is the new way to discover new targeting molecules and in particular new peptides. The possibility of being able to chemically modify them quickly and easily is certainly an advantage. For this reason, we have described in this review the most recent and most used chemical modifications that can be made in order to make peptides both more effective and safer for the treatment of human diseases. However, much remains to understand how to improve the binding affinity and thus reduce the side effects of numerous treatments in patients.

## Figures and Tables

**Figure 1 pharmaceutics-15-01648-f001:**
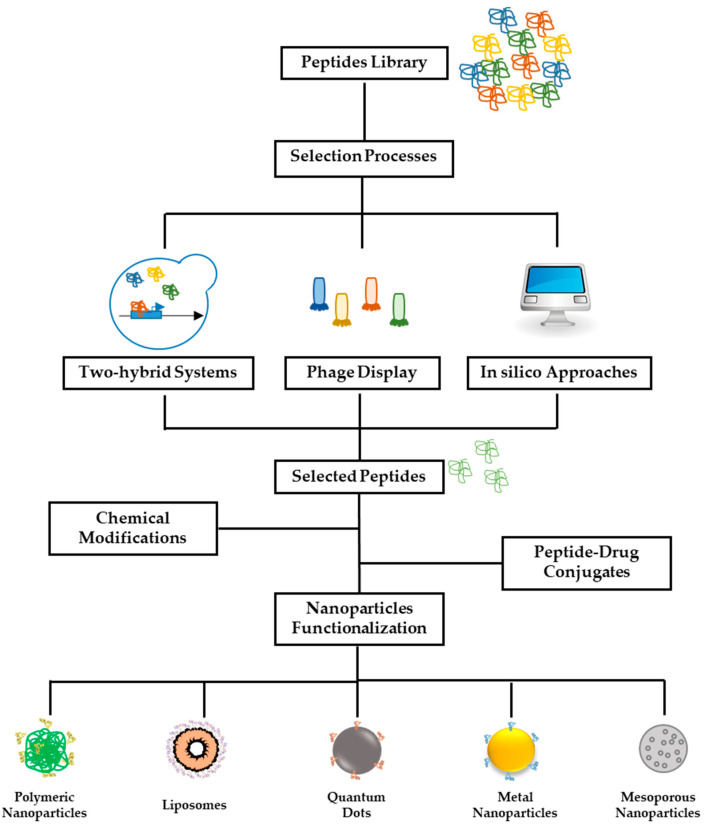
Schematic representation of selection processes for peptides. Starting from a library with thousands of different clones, different methods can be used for the identification of suitable sequences able to recognize specific targets. In particular, two-hybrid and phage display techniques are defined as in vivo techniques, while in silico analysis is based on computational design and molecular dynamics simulation systems.

**Figure 2 pharmaceutics-15-01648-f002:**
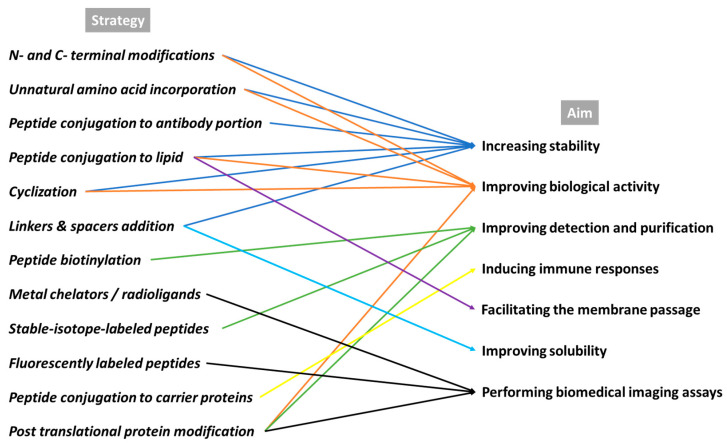
Overview of the main chemical modifications designed to improve in vivo peptide features.

**Figure 3 pharmaceutics-15-01648-f003:**
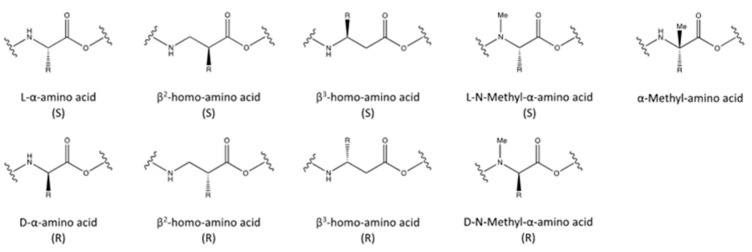
Chemical structures of α-amino acids, β^2^- and β^3^-homo-amino acids, N-methyl-α-amino acids, and α-Methyl-amino acids in both (S)- and (R)-stereochemistry.

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
