# Peer review of "Targeting Peptides: The New Generation of Targeted Drug Delivery Systems"

_pharmaceutics, 2023, doi:10.3390/pharmaceutics15061648_

Round 1
Reviewer 1 Report
Comments to authors:
1. Instead of saying "different disorders," the author should give more particular instances of diseases that peptides have been explored for.
2. More detail on the methods of experimental and in silico design, as well as the chemical modifications that improve the stability and efficacy of peptides, would be helpful.
3. To help readers better grasp the potential effect of this research, the author should include more information about how the use of peptides could help overcome physiological difficulties and improve existing treatments.
4. An odd phrase like "they found application in numerous research sectors where they were widely utilized, especially in cancer treatment" can be found in the introduction. There is room for improvement in the clarity and conciseness of this sentence.
5. It would be helpful to have a more in-depth discussion of the pros and cons of using peptides as targeting moieties, as opposed to alternative possibilities like antibodies or oligonucleotide aptamers, in the introduction. Readers would benefit from a more in-depth examination of the benefits and drawbacks of each alternative so that they may make an informed decision when choosing a targeted moiety for a certain application.
6. additional commentary on the difficulties that may arise when employing peptides as targeting moieties. While some of the caveats of using other targeting moieties like antibodies are mentioned, the difficulties and restrictions of working with peptides are not addressed in the same depth. To better grasp the possibilities of this technique, it would be helpful to explore both the benefits and drawbacks of using peptides as targeting moieties.
7. More information on the drawbacks and potential problems associated with two-hybrid systems, such as high false positive and negative rates and toxicity to yeast cells, would be helpful in the section on this topic.
8. More detail on the multi-step selection and enrichment process that is bio-panning would be helpful in the phage display section.
9. The section on in silico approaches, which explains the various algorithms used in peptide selection, is too brief.
10. More specific examples of peptides with short half-lives in plasma and how specific modifications have been shown to increase stability and prolong shelf life could be provided in the chemical conjugation section.
11. Despite mentioning that N- and C-terminal modifications can improve stability and solubility, the chemical conjugation doesn't address the drawbacks that might come with them, like reduced bioactivity or a changed conformation.
12. More context regarding the significance and utility of peptide labeling would enhance the manuscript. This may aid the reader in appreciating the value of the various labeling strategies.
13. Some of the claims made in Section 4 are unsupported by references. For example, the claim that PDCs are less toxic because they are quickly eliminated by the kidneys More information about the various linkers used in PDCs and their respective applications would be helpful in this section as well.
14. Liposomes are used as medication delivery vehicles, including how various compounds are loaded into the liposomes and how their properties can be modified for diverse uses. The notion that painting liposomes with exogenous peptides is a frequent practice, for example, is not backed up by any references.
15. Targeted drug delivery using nanocarriers like quantum dots (QDs) and mesoporous silica nanoparticles (MSNs) is discussed in detail. The language and writing style, however, may be tightened up for more clarity.
16. In this section, you'll learn about QDs and how they can be used in the imaging and diagnosis of cancer. However, more information on QDs, including their synthesis and characterization, would enrich the section.
17. The discussion of MSNs gives a helpful summary of the substance classes, their characteristics, and their applications in drug delivery. However, more details on the synthesis and characterization of MSNs, as well as their benefits and drawbacks in comparison to other types of nanocarriers, would make this section more useful.
18. The limitations and possible benefits of targeting peptides are well summarized at the end, but a more in-depth examination of where this research is headed would be helpful.
Author Response
Dear Editor Dr. Anthea Wang,
We would like to take this chance to thank you and the Reviewers for the consideration and the time devoted to our manuscript pharmaceutics-2400197. Referees have raised some interesting concerns, and we have revised our manuscript accordingly for your further consideration. All alterations in the manuscript are marked with the “track changes”, and our point-by-point response is in the following.
We thank you and the Referees for the helpful and insightful comments, which we believe helped to strengthen the quality of the manuscript.
In particular, we revised the entire manuscript for typos and grammar errors, we improved all the sections of the work accordingly to reviewer suggestions.
Sincerely yours,
On behalf of all co-Authors,
Dr. Melissa Santi Reviewer 1: Answer: We thank the Reviewer for the valuable comments, and we guess her/his concerns are addressed in the following point-by-point response. 1. Instead of saying "different disorders," the author should give more particular instances of diseases that peptides have been explored for. Answer: We thank the reviewer for his/her suggestion we added some example of peptides used in preclinical research in the Introduction (line 65-79) 2. More detail on the methods of experimental and in silico design, as well as the chemical modifications that improve the stability and efficacy of peptides, would be helpful. Answer: We thank the Reviewer for his/her suggestion, we improved the introduction and all the selection processes. 3. To help readers better grasp the potential effect of this research, the author should include more information about how the use of peptides could help overcome physiological difficulties and improve existing treatments. Answer: We thank the Reviewer for his/her suggestion, however in each section examples about the use and improvements of targeting peptides are already included. 4. An odd phrase like "they found application in numerous research sectors where they were widely utilized, especially in cancer treatment" can be found in the introduction. There is room for improvement in the clarity and conciseness of this sentence. Answer: We thank the reviewer for the careful check, we changed the phrase to make it more clear (now lines 24-26). 5. It would be helpful to have a more in-depth discussion of the pros and cons of using peptides as targeting moieties, as opposed to alternative possibilities like antibodies or oligonucleotide aptamers, in the introduction. Readers would benefit from a more in-depth examination of the benefits and drawbacks of each alternative so that they may make an informed decision when choosing a targeted moiety for a certain application. 6. additional commentary on the difficulties that may arise when employing peptides as targeting moieties. While some of the caveats of using other targeting moieties like antibodies are mentioned,
the difficulties and restrictions of working with peptides are not addressed in the same depth. To better grasp the possibilities of this technique, it would be helpful to explore both the benefits and drawbacks of using peptides as targeting moieties. Answer to 5 and 6: We thank the reviewer for his/her suggestion. We improved the Introduction with advantages in using targeting peptides instead of antibodies or oligonucleotides. The major cons in using peptides is represented by their lower affinity for the targets, and we already described this point in the introduction. Some of the drawbacks for the use of peptides were also mentioned at the beginning of Chapter 3. 7. More information on the drawbacks and potential problems associated with two-hybrid systems, such as high false positive and negative rates and toxicity to yeast cells, would be helpful in the section on this topic. Answer: We thank the reviewer for his/her suggestion. We better described two-hybrid system and discussed troubleshooting in section 2.1.1 8. More detail on the multi-step selection and enrichment process that is bio-panning would be helpful in the phage display section. Answer: We thank the reviewer for his/her suggestion. We better described the bio-panning process in section 2.1.2 9. The section on in silico approaches, which explains the various algorithms used in peptide selection, is too brief. Answer: We thank the reviewer for his/her suggestion. The in silico paragraph was extended. 10. More specific examples of peptides with short half-lives in plasma and how specific modifications have been shown to increase stability and prolong shelf life could be provided in the chemical conjugation section. Answer: We thank the reviewer for his/her suggestion. We added some examples about N-terminal modification, unnatural amino acid incorporation and cyclization, in which substance P, neurotensin, octreotide and cRGD peptides were applied. 11. Despite mentioning that N- and C-terminal modifications can improve stability and solubility, the chemical conjugation doesn't address the drawbacks that might come with them, like reduced bioactivity or a changed conformation. Answer: We thank the reviewer for his/her suggestion. We added a sentence addressing these possible drawbacks. 12. More context regarding the significance and utility of peptide labeling would enhance the manuscript. This may aid the reader in appreciating the value of the various labeling strategies. Answer: We thank the reviewer for his/her suggestion. We added the sentence “peptides that are generally quickly metabolized (e.g. in liver) and excreted (e.g. from kidneys) need chemical modifications extending the biological half-life also to meet the imaging requirement of the radiolabeled peptides” in the introduction of section 3 as to guide the reader towards the labelling 3.8, 3.9 and 3.10 sections. Moreover, an example regarding the fluorescent 18F-labeled vasoactive intestinal peptide (VIP) was added. 13. Some of the claims made in Section 4 are unsupported by references. For example, the claim that PDCs are less toxic because they are quickly eliminated by the kidneys More information about the various linkers used in PDCs and their respective applications would be helpful in this section as well.
Answer: We added some references and improved PDCs section. 14. Liposomes are used as medication delivery vehicles, including how various compounds are loaded into the liposomes and how their properties can be modified for diverse uses. The notion that painting liposomes with exogenous peptides is a frequent practice, for example, is not backed up by any references. Answer: We thank the reviewer for his/her careful check, we added a reference as suggested. 15. Targeted drug delivery using nanocarriers like quantum dots (QDs) and mesoporous silica nanoparticles (MSNs) is discussed in detail. The language and writing style, however, may be tightened up for more clarity. Answer: We thank the reviewer for the suggestion we improved these two paragraphs and check for typos and errors to make it clearer. 16. In this section, you'll learn about QDs and how they can be used in the imaging and diagnosis of cancer. However, more information on QDs, including their synthesis and characterization, would enrich the section. Answer: We improved the QD section with some details about the synthesis process. 17. The discussion of MSNs gives a helpful summary of the substance classes, their characteristics, and their applications in drug delivery. However, more details on the synthesis and characterization of MSNs, as well as their benefits and drawbacks in comparison to other types of nanocarriers, would make this section more useful. Answer: MSNs section was improved as suggested 18. The limitations and possible benefits of targeting peptides are well summarized at the end, but a more in-depth examination of where this research is headed would be helpful. Answer: We improved the conclusions. Reviewer 2: The review presented by Todaro et al. on "Targeting peptides: the new generation of targeted drug delivery systems" described the latest technologies on the peptide targets and their applications in the development of targeted drug delivery systems. The review covers all aspects and written well. However few suggestions has to be implicated in the revision. Answer: We thank the Reviewer for the valuable comments, and we guess her/his concerns are addressed in the following point-by-point response. Comments 1. Section 2.1.2. Phage display can be improved well Answer: We thank the reviewer for his/her suggestion. We better described the bio-panning process in section 2.1.2 2. Section - 2.2.2 In silico methods - this section should be added with the tools used and their websites for easy access. Answer: We thank the reviewer for his/her suggestion. The in silico paragraph was expanded.
3. A diagrammatic representation for the Peptide-based targeting strategies for drug-delivery systems would be easy to understand for the authors. Answer: We agree with the Reviewer that a diagram that better explain the possible peptide-based targeting strategies for drug delivery system will be very helpful for the readers. The figure 1 was modified with all steps involved in the formulation of new targeted drug delivery systems. Minor comments: 1. Line no.48 - remove the "r". Also keep the full stop after quoting the references. 2. Line no. 111- sequences are isolated amplified - can be changed to " sequences are isolated, amplified or isolated followed by amplified 3. Line no. 123- Correct it to Phage Display 4. Line no.130 - remove the "s". 5. check the formats in section 2.2 and 2.3 6. Avoid all typographical and linguistic errors during the revision Answer: We thank the reviewer for his/her careful check. We made all the suggested changes and checked all the work to correct errors. Reviewer 3: The authors gave a comprehensive overview of the recent progress in developing targeting peptides. The work is well supported, and the evidence is laid out quite clearly. It will be a very helpful summary for the researchers in the field. Answer: We thank the Reviewer for the valuable comment and to spent some time to read our work. We really appreciate his/her interest and feedback.

Reviewer 2 Report
The review presented by Todaro et al. on "Targeting peptides: the new generation of targeted drug delivery systems" described the latest technologies on the peptide targets and their applications in the development of targeted drug delivery systems. The review covers all aspects and written well. However few suggestions has to be implicated in the revision.
Comments
1. Section 2.1.2. Phage display can be improved well
2. Section - 2.2.2 In silico methods - this section should be added with the tools used and their websites for easy access.
3. A diagrammatic representation for the Peptide-based targeting strategies for drug-delivery systems would be easy to understand for the authors.
Minor comments:
1. Line no.48 - remove the "r". Also keep the full stop after quoting the references.
2. Line no. 111- sequences are isolated amplified - can be changed to " sequences are isolated, amplified or isolated followed by amplified
3. Line no. 123- Correct it to Phage Display
4. Line no.130 - remove the "s".
5. check the formats in section 2.2 and 2.3
6. Avoid all typographical and linguistic errors during the revision
Author Response

(The authors gave the same response as above.)

Reviewer 3 Report
The authors gave a comprehensive overview of the recent progress in developing targeting peptides. The work is well supported, and the evidence is laid out quite clearly. It will be a very helpful summary for the researchers in the field.
Author Response

(The authors gave the same response as above.)

Round 2
Reviewer 1 Report
No further comments to authors
Reviewer 2 Report
Authors implemented all the corrections as suggested in the queries during the revision and hence it can be accepted